# Human Papillomavirus and Associated Cancers: A Review

**DOI:** 10.3390/v16050680

**Published:** 2024-04-26

**Authors:** JaNiese E. Jensen, Greta L. Becker, J. Brooks Jackson, Mary B. Rysavy

**Affiliations:** 1Department of Pathology, University of Iowa Carver College of Medicine, Iowa City, IA 52242, USA; janiese-jensen@uiowa.edu (J.E.J.); greta-becker@uiowa.edu (G.L.B.); brooks-jackson@uiowa.edu (J.B.J.); 2Department of Obstetrics, Gynecology & Reproductive Sciences, McGovern Medical School, University of Texas Health Science Center at Houston, Houston, TX 78030, USA

**Keywords:** human papillomavirus, cervical cancer, HPV vaccine, HIV

## Abstract

The human papillomavirus is the most common sexually transmitted infection in the world. Most HPV infections clear spontaneously within 2 years of infection; however, persistent infection can result in a wide array of diseases, ranging from genital warts to cancer. Most cases of cervical, anal, and oropharyngeal cancers are due to HPV infection, with cervical cancer being one of the leading causes of cancer death in women worldwide. Screening is available for HPV and cervical cancer, but is not available everywhere, particularly in lower-resource settings. HPV infection disproportionally affects individuals living with HIV, resulting in decreased clearance, increased development of cancer, and increased mortality. The development of the HPV vaccine has shown a drastic decrease in HPV-related diseases. The vaccine prevents cervical cancer with near 100% efficacy, if given prior to first sexual activity. Vaccination uptake remains low worldwide due to a lack of access and limited knowledge of HPV. Increasing awareness of HPV and access to vaccination are necessary to decrease cancer and HPV-related morbidity and mortality worldwide.

## 1. Introduction

The human papillomavirus (HPV) is a nonenveloped, double-stranded DNA virus, and the most common sexually transmitted infection worldwide. The icosahedral-shaped virus was first described using electron microscopy in 1949 by Strauss et al. [1,2]. In 1976, Harald zur Hausen postulated a link between HPV and cervical cancer, and within the following decade, his team successfully isolated two novel strains, HPV 16 and 18, from cervical cancer biopsies, experiments that would later earn him a Nobel Prize in medicine [2]. Today, more than 170 genotypes of HPV, categorized as low-risk and high-risk genotypes, have been described [3]. Among them, there are at least 12 high-risk strains, with HPV 16, 18, 31, and 45 responsible for most HPV-causing cancers [4]. Low-risk strains, such as HPV 6 and 11, rarely lead to cancer, but can cause warts of the genitals, anus, mouth, and throat [5].

The mechanism of HPV infection is not fully understood. The most commonly accepted mechanism is that HPV enters cells through microlesions in the epithelial basement membrane via endocytosis, then is translocated to the nucleus for genome replication and transcription [6]. The genome of HPV has eight opening reading frames: E1, E2, E4, E5, E6, and E7; and two late regions: L1 and L2. The reading frames facilitate replication and transcription, processes completed by host proteins. Research has primarily focused on frames E6 and E7 due to their role in disrupting cell growth and differentiation. The late regions, L1 and L2, function to package the amplified genome into a virion creating the icosahedral capsid. Eventually, the virus exits cells through natural desquamation, commonly infecting epithelial cells.

HPV is transmitted through sexual contact, commonly via skin–skin or skin–mucosal contact [7]. More than 80% of sexually active adults will be exposed to HPV in their lifetimes [8]. Much more rarely, it can be spread vertically during the perinatal period. There have also been rare reports of autoinoculation or indirect infection in individuals who have never had any sexual contact [7].

## 2. Disease and Prevalence

HPV infection is most commonly asymptomatic. Low-risk strains of HPV can cause plantar, common, or genital warts, and focal respiratory papillomatosis, while high-risk strains are associated with cervical intraepithelial neoplasia (CIN) and cervical, vaginal, vulvar, anal, penile, and oropharyngeal carcinomas [5]. It is estimated that 90% of cervical and anal cancers, 70% of vulvar and vaginal cancers, 60% of penile cancers, and 70% of oropharyngeal cancers are due to persistent, high-risk HPV infection, as depicted in Figure 1 [9,10].

HPV infection causes cancer due to upregulation of E6 and E7 reading frames in the HPV genome which inactivate p53 and Rb, respectively, in their host cells [11]. The upregulation of E6 and E7 can lead to an abortive infection in which there is dysregulation of the viral genome preventing the production of infectious viruses. HPV can also lead to productive infections that allow for the viral life cycle to function properly. The specific mechanism of carcinogenesis after HPV infection varies based on the location of the infection. Cervical cancer commonly develops where the cervix transitions from stratified squamous epithelium to columnar epithelium known as the transformation zone. This area is postulated to consist of reserve cells that act as stem cells. Once infected with HPV, the viral gene expression cannot be regulated, leading to metaplasia and eventually cancer. The dysregulation of the viral life cycle in reserve cells is due to an increase in abortive infections rather than productive infections. Anal cancer due to HPV infection occurs in a similar pattern to cervical cancer as HPV often infects the anal transformation zone, leading to cancer. Both cervical and anal HPV carcinogenesis differ from the likely mechanism of HPV-associated oropharyngeal cancer. The oropharynx is made up of tonsillar crypt epithelium which more readily allows HPV infection that is more permissive of abortive infections than productive infections. This increases the risk of dysregulated viral gene expression that can lead to cancer.

According to the 2022 Global Cancer Observatory from the World Health Organization (WHO), cervical cancer was the eighth most commonly diagnosed cancer and the ninth leading cause of cancer-related deaths [12]. Cervical cancer is the second most common cancer in African and Southeast Asian females, with breast cancer being the most common [12]. Oropharyngeal cancer is the 24th most common cancer worldwide, with an incidence of 106,400 cases [12]. Of those cases, an estimated 70% are caused by high-risk HPV infection [10].

Screening for HPV infections is typically only completed for women and is not universally performed; thus, true prevalence is difficult to obtain. In low-to-middle-income countries (LMICs), the screening rate for cervical cancer is estimated at 44% [13]. In the United States, screening is estimated at 72.4% [14]. A study of 1892 patients in western Uganda in 2019 found the prevalence of high-risk HPV was 21% in women aged 25 to 49 years from self-collected samples [15]. In China, the prevalence of cervical HPV in a study of 427,401 women aged 20 years or older in 2021 was 15.0% and high-risk HPV was 12.1% [16]. In the United States, overall HPV prevalence was 40.0% for men and women aged 15 to 59 years, with a high-risk prevalence of 24.2% and 19.9% in men and women, respectively [17]. Males are not frequently screened for HPV. However, a recent meta-analysis estimated that one in three males over the age of 15 are or have been infected with HPV, and 21% were infected with high-risk HPV [18].

## 3. Diagnosis and Prognosis

Screening is most commonly performed through HPV co-testing during a cervical Papanicolaou (Pap) test in the United States and many other higher-resource settings. According to the United States Preventative Services Task Force (USPSTF) and American Society for Colposcopy and Cervical Pathology recommendations, Pap tests are recommended every three years for women beginning at age 21. For screening in women older than 30 years, women may choose to have HPV co-testing administered with a cervical Pap test every five years, HPV testing alone every five years, or Pap testing every three years, all with completion at age 65 if previous testing is negative [19]. The CDC recommends screening women living with HIV yearly for three years and then every three years for life with a Pap test or Pap test with HPV co-testing starting at age 21 [20]. If both cervical Pap test and HPV testing are negative, the incidence of developing CIN III or greater within five years is 0.7% [21]. As of 2021, the WHO recommends screening with HPV DNA testing only, rather than Pap test cytology or visual inspection with acetic acid (VIA), for all women starting at age 30 with an interval of every five to ten years with completion after two negative screening tests [22]. This recommendation changes for women living with HIV. Initial screening with HPV DNA testing starts at age 25 at an interval of three to five years [22]. The comparison of guidelines is shown in Table 1. A study of 12,113 women in the Netherlands found the prevalence of high-risk HPV to be 8% when screening with only HPV DNA testing [23]. HPV DNA screening of 2453 Iranian women showed 10.3% had high-risk HPV [24]. The WHO notes that HPV DNA screening allows for self-sampling and could be used to increase screening rates as some women may feel more comfortable with self-sampling than with traditional sampling with a provider [22].

There are many laboratory methods to test for HPV infection. Co-testing in the United States is commonly completed during a Pap test, in which pathologists evaluate a cytology smear of cells from the uterine cervix, and signal amplification assays such as the Digene^®^ HPV test or Cervista^®^ HPV high-risk assay [25]. Polymerase chain reaction (PCR) testing and viral load quantification may also be used to test for HPV [25]. Screening for HPV infection in men is not currently recommended by the Centers for Disease Control and Prevention (CDC) as there are no approved tests currently; however, anal Pap testing may be offered to males who receive anal sex or are living with human immunodeficiency virus (HIV) [26].

Because Pap screening leads to a multistep process that requires pathology equipment and professionals for diagnosis and treatment of cervical dysplasia or cancer, many lower resource settings use visual inspection with acetic acid (VIA). VIA is completed with 3–5% acetic acid, which is applied to the cervix for 1 to 2 min [27]. The cervix is then observed for a white color change, which, if present, would indicate potential for underlying pathology. A 2015 study of 39,740 women that analyzed the diagnostic accuracy of VIA showed that 7.1% of screened women had a positive VIA and, of those, pathology confirmed 92.8% had a normal cervix or lesion of CIN I, indicating a low positive predictive value of VIA screening [28].

Screening for HPV is primarily focused on patients who are considered high-risk. This means screening is often for cervical and anal cancers, with limited screening options for HPV-associated oropharyngeal cancers. A potential solution to this is screening with a plasma-circulating marker for HPV known as ctHPVDNA. In a study of 97 patients with locally confined HPV-associated oropharyngeal squamous cell carcinoma, HPV16 ctDNA was detected in 90 patients, and HPV33 ctDNA was detected in three patients [29]. This shows that the ctHPVDNA test is 95.6% sensitive. All controls had undetectable levels, and therefore a 100% specificity [29]. Although this test is primarily being used in research settings, the high sensitivity and specificity show promise for use as a screening test, particularly to screen for oropharyngeal cancers related to HPV.

Overall, the prognosis of HPV infection itself remains good. Within 12–24 months, 90% of infections are undetectable, suggesting spontaneous clearance of the virus [30]. Uncleared infections can cause clinical disease. Mortality due to HPV is most commonly due to cervical cancer. In an analysis of cervical cancer of all stages in the United States from 2013 to 2019, there was a 67.2% 5-year survival rate for all individuals [31]. If the cancer had metastasized, the 5-year survival rate decreased to 19% [31]. A 2022 study analyzing ICD-10 codes showed that overall survival for individuals who had HPV-associated oropharyngeal cancer was 81.5% in males and 80.6% in females [32].

## 4. Current Treatments

There are currently no well-studied treatments for the human papillomavirus infection. The treatments available target the symptoms and diseases caused by HPV infection such as warts or cancer.

For genital warts, treatments include topical therapies and physical removal or destruction. The topical therapies include imiquimod, podophyllotoxin, sinecatechins, and isotretinoin, with the most commonly used options being imiquimod and podophyllotoxin [33]. The physical removal and destruction treatments include simple surgical excision, liquid nitrogen ablation, electrocauterization, and photodynamic therapy [33].

The options for treatment of CIN II or III include conization, cryotherapy, and loop electrosurgical excision procedure (LEEP) [34]. For cervical cancer, staging remains important to determine treatment. Due to the significantly higher burden of cervical cancer in LMICs, historically, the staging system for cervical cancer was clinically based. This system allowed countries, particularly LMICs, to better diagnose and stage cervical cancer based on available resources. In 2018, the International Federation of Gynecology and Obstetrics (FIGO) incorporated cross-sectional imaging and pathology into the staging system [35]. Earlier stages may be treated with simple hysterectomy or conization, while later stages may require hysterectomy, radiation, and/or chemotherapy [34]. Chemotherapy has historically been cisplatin-based; however, with an increase in resistance to monotherapy, treatment has been more efficacious when cisplatin-based chemotherapy is combined with another therapy such as topotecan, paclitaxel, 5-fluorouracil, or bleomycin [34]. For locally advanced cervical cancer, radiation is often recommended alongside the combined chemotherapy regimen [34].

As with cervical cancer, anal cancer treatments rely on staging. Local excision is available for small lesions that do not involve sphincters and often do not require further treatment [36]. Patients with more advanced disease require treatment prior to excision, which is commonly radiation and chemotherapy. The most commonly used multimodal approach is local excision followed by mitomycin, 5-fluorouracil, and radiation. If the cancer is metastatic, the addition of a cisplatin-based therapy is necessary.

HPV-associated oropharyngeal cancer follows a similar treatment pattern to other HPV-associated cancers [37]. Excision is generally the first treatment and can be completed either with an open procedure or less invasive options of laser and robotic surgery. The standard of care following excision is radiation with cisplatin-based chemotherapy.

## 5. HPV Prevention

Methods of preventing HPV infection are targeted at preventing exposure to the virus. Sexual activity is the most frequent mode of transmission of HPV [7]. Primary prevention of HPV can be achieved through abstinence or vaccination prior to sexual activity. One study found that women aged 35–60 with five or more lifetime sexual partners had a greater than two-fold increase in the detection of high-risk HPV as compared to those with less than five lifetime sexual partners [38]. In a meta-analysis of HPV infections and condom use, all eight studies showed some protective effect of condom use, and four of the eight showed a very significant protective effect in the prevention of HPV [39]. Regarding vertical transmission, one study found that 5.2% of neonates with a mother who had a positive HPV test during pregnancy also had detectable HPV [40]. With low vertical transmission rates, cesarean section is not recommended if the mother is infected with HPV. A 2016 study found that perinatal transmission of the virus was decreased by 46% with cesarean section; however, transmission occurred in 15% of cesarean sections [41]. The best way to prevent vertical transmission from mother to child is by protecting the mother from becoming infected with HPV. To prevent autoinoculation and indirect contact, good hand hygiene remains important.

## 6. HPV Vaccines and Efficacy

The HPV vaccine was first approved for use in the United States in 2006 [42]. There are currently three major types of vaccines available: bivalent, quadrivalent, and nonavalent [42]. The bivalent vaccine targets HPV 16 and 18, the quadrivalent vaccine targets HPV 6, 11, 16, and 18, and the most recent vaccine is the nonavalent, or 9-valent, vaccine under the Merck & Company brand-name Gardasil^®^, which targets HPV 6, 11, 16, 18, 31, 33, 45, 52, and 58 [42].

The HPV vaccine is a non-infectious recombinant vaccine that is created from virus-like particles of the L1 protein which is purified from each of the associated HPV types [43]. An antibody response is expected one month after completing the vaccine series in over 98% of people. No minimum protective antibody titer level has been identified [43]. The vaccine also increases B cell immunity by changing the type of antibodies in circulation. In response to a natural HPV infection, non-neutralizing antibodies are formed [44]. If one dose of the vaccine is given after infection with HPV 16, neutralizing antibodies are formed [44]. The CDC notes that the HPV vaccine provides long-lasting protection against infection with no waning immunity for at least 12 years [45].

The efficacy of the HPV vaccine can be assessed by its ability to prevent low-risk and high-risk HPV infections and diseases. In the prevention of genital warts, the quadrivalent vaccine showed a significant reduction in the risk of genital warts (OR = 0.03, 95% CI 0.01–0.09) and a significantly reduced number of genital warts (OR = 0.36, 95% CI 0.26–0.51) [46]. In the prevention of cervical cancer, a study in Scotland by Palmer et al. analyzed cancer registries and vaccine records of women born from 1988–1996 to assess for a connection between the HPV vaccine and cervical cancer [47]. This study showed that women who were vaccinated at age 12–13 years with the bivalent HPV vaccine had no cases of invasive cervical cancer, indicating 100% vaccine efficacy [47]. In women aged 14–22 years who received three doses of the vaccine, 3.2 out of 100,000 had cervical cancer as compared to 8.4 out of 100,000 in unvaccinated women [47]. This study shows that early vaccination for the human papillomavirus is effective in preventing cervical cancer and is most effective when given to girls 13 years or younger.

The vaccines have also been shown to reduce oropharyngeal and anal infections and thus prevent subsequent cancer. A 2021 systematic review found that those who were vaccinated had an 82.7% relative reduction percentage of oral and oropharyngeal HPV infections [48]. For anal HPV infections, a meta-analysis found the incidence of anal cancer to be significantly reduced after HPV vaccination when compared to a control group (RR = 0.42, 95% CI 0.31–0.57; *p* = 0.02) [49].

## 7. Vaccination Recommendations and Uptake

The CDC recommends two doses of the HPV vaccine for 11- and 12-year-old children, both girls and boys, with each dose spaced six to twelve months apart [42]. Individuals aged 15–26 years, who have not been vaccinated previously, should receive three doses of the HPV vaccine [42]. However, based on factors such as early sexual debut or immunocompromised status, the CDC suggests vaccination as early as 9 years, with catch-up vaccination up to 45 years [42].

The WHO recommends one or two doses for girls aged 9–14 years and women aged 15–20, and two doses with a 6-month interval for women older than 21 [50]. If a person is immunocompromised or living with HIV, the WHO recommends at least two doses, with three doses if feasible [50]. The age range for HPV vaccination allows for flexibility in clinical practice. However, the vaccine is most efficacious if given prior to sexual debut [51]. Therefore, risk assessment and shared decision-making should be performed for all patients to determine the best age to complete the vaccination series.

The vaccine is not available in every country in the world. As of June 2020, 107 of 194 WHO member states had introduced at least one type of HPV vaccine [52]. The percentage of countries in the Americas, Europe, Oceania, Asia, and Africa that have HPV vaccinations available is 85%, 77%, 56%, 40%, and 31%, respectively, as displayed in Figure 2 [52]. It is hard to determine worldwide uptake of the vaccine due to variability in availability, which is particularly low in African and Oceanic countries. Global estimates suggest the rate of completion of the vaccine series is 15% [52]. Uptake of the vaccine was 19.6% in girls aged 9–14 in northern Uganda [53], 24.4% in general uptake in China [54], and 58.5% in girls aged 13–15 in the United States [55]. These data suggest that the economic status of the country may influence vaccine uptake.

A 2017 CDC survey looked at factors associated with HPV vaccine uptake. Adolescents from low-income families were more likely to receive the vaccine compared to higher-income families (OR = 1.21; 95% CI 1.04–1.43) [56]. This study also found that males (OR = 0.74; 95% CI 0.66–0.83) and adolescents with a family physician not suggesting vaccination (OR = 0.57; 95% CI 0.49–0.05) were less likely to receive the vaccine, and adolescents who did not have safety concerns about the vaccine were more likely to receive the vaccine (OR = 3.24; 95% CI 2.68–3.93) [56]. Acknowledging that not every country has access to the vaccine, some ways to increase uptake may include school campaigns, sharing information on the efficacy and safety of the vaccine, advocacy for more countries to add the HPV vaccine to recommended vaccination schedules, and policy change to allow or recommend that boys and men also be vaccinated throughout the world.

## 8. Co-Infection of HPV and HIV

Human immunodeficiency virus (HIV) is a virus that targets human helper T cells, known as CD4 T cells, and can lead to immunodeficiency as the virus continues to attack T cells [57]. The last, and most severe stage, of HIV infection is acquired immunodeficiency syndrome (AIDS) [58]. This occurs when the patient’s CD4 T cell count falls below 200 cells per milliliter of blood or if a patient develops opportunistic infections [58]. HIV affects 39 million people globally and is most prevalent in African countries with 1 in every 25 African adults living with HIV [59]. It is well known that HIV can cause an increased risk of infections and cancers. The connection between HIV and HPV infection is well known, and invasive cervical carcinoma is considered an AIDS-defining illness [60]. Patients living with HIV have twice the rate of HPV acquisition and half the HPV clearance rate as compared to those without HIV [61]. In a Ugandan study, 49% of postpartum women were infected with a high-risk HPV genotype, and women living with HIV had an HPV prevalence of 86% compared to 59% in women without HIV (*p* = 0.003) [62]. Those living with HIV have a decreased likelihood of clearing HPV infections when compared to those without HIV (HR = 0.31; 95% CI 0.21–0.45) [63]. With decreased clearance of HPV, particularly high-risk HPV, there is an increased risk of progression to cancer. Of all new cervical cancer diagnoses worldwide in 2018, 5.8% were in women living with HIV [64]. The greatest connection between HIV and cervical cancer was seen in Africa. Twenty-five percent of African women with newly diagnosed cervical cancer also had HIV [64]. The association was significantly lower in other regions with 1.7% in the Americas, 1.5% in Europe, and 1.4% in Southeast Asia [64]. Patients with invasive cervical cancer and HIV have an increased risk of death from cancer (HR = 1.95; 95% CI 1.20 to 3.17) [65]. Due to the increased risk of co-infection, the WHO recommends that patients living with HIV should complete the three-series HPV vaccination [50]. Most recommendations also include increased screening for HPV in patients living with HIV as well.

## 9. Prospective Therapeutic Use of HPV Vaccines

Advancements in therapeutic strategies offer promise for the treatment of HPV infection with the hope of preventing its associated diseases. One avenue for this is the development of therapeutic vaccines. These are entirely new medications and similar to currently available vaccines only insofar as they are delivered via injection. A systematic review and meta-analysis completed in 2023 by Ibrahim Khalil et al. analyzed the effectiveness of a variety of newly developed therapeutic HPV vaccines in treating cervical cancer [66]. The therapeutic vaccines used in this study were peptide-based, protein-based, viral-vectored, bacterial-vectored, DNA-based, and cell-based vaccines [66]. The targets of many therapeutic vaccines are E6 and E7 proteins, which are the main proteins that influence infection and carcinogenesis [66]. It has been postulated that targeting the E2 protein could be an ideal target for therapeutic vaccines as the E2 protein acts as a repressor of E6 and E7 proteins and could therefore prevent dysregulation of p53 and Rb [67]. This analysis found that if a therapeutic vaccine is given to a patient with CIN II or III, the pooled outcome was a 54% regression of the lesions to normal or to CIN I [66]. Comparatively, patients who had CIN II or III and were given a placebo vaccine had a regression of 27% [66]. One study from the meta-analysis looked at a therapeutic vaccine only targeting the E2 protein and showed complete clearance of lesions in individuals with CIN I, II, or III lesions [68]. There are currently multiple therapeutic vaccines undergoing phase II and III trials, none of which have yet received approval for clinical use [69]. However, these findings suggest that utilizing a therapeutic HPV vaccine as treatment for CIN may be feasible and may help lower rates of progression to cervical cancer and associated deaths.

Some investigators have also looked at the use of the widely available prophylactic HPV vaccine as a potential treatment for HPV-associated disease. Prophylactic vaccines differ from therapeutic vaccines as described above as prophylactic vaccines are typically recombinant vaccines created from viral-like particles [43]. A randomized control trial in Iran by Karimi–Zarchi et al. analyzed the effect of the currently available prophylactic quadrivalent HPV vaccine as a therapeutic agent [70]. This study showed regression of CIN I, II, and III lesions in those who had received two or more doses of the vaccine at a rate of 75.6%, 78%, and 72.1%, respectively [70]. These regression rates can be compared to the control group which had rates of regression of 45.7% for CIN I, 40% for CIN II, and 41.2% for CIN III lesions [70]. The regression rates were significantly reduced for all three grades when comparing those who received the vaccine and those who did not (*p* = 0.02, 0.03, 0.03) [70]. A 2020 meta-analysis of prophylactic vaccines by Jentschke et al. reported on the importance of removal of the cervical lesion by a conization procedure prior to vaccination for risk reduction of CIN lesions [71]. This study found a 3.1% risk of recurrence of CIN II or greater after conization and subsequent HPV vaccination as compared to a 5.3% risk of recurrence in those who had conization but had not received the vaccine [71].

The need for surgical resection prior to vaccination is likely explained by a decrease in TNF-alpha and pro-inflammatory cytokines at the cervix which lowers the likelihood of HPV infection persistence, thus making the target of HPV vaccinations similar to a naïve environment [71]. The vaccines are in these cases essentially preventing reinfection, rather than actually treating it, but nonetheless did seem to have a lower risk of recurrence of CIN. A 2023 review also discussed the potential mechanism of prophylactic vaccination, noting that post-resection HPV-associated disease requires new infection of noninfected cells which can be prevented with HPV vaccination [72]. This analysis emphasized that while the vaccine prevents new infection or re-infection of HPV, it remains unlikely that a prophylactic vaccine would prevent disease progression if the residual disease remains [72].

Although currently available prophylactic HPV vaccines and newly developed therapeutic HPV vaccines have shown promise in the treatment and prevention of HPV-associated disease, further research is required to determine the efficacy of each vaccine type. This need for additional data is reflected in clinical guidelines, as they have yet to be changed to include any recommendations regarding prophylactic and therapeutic HPV vaccine use.

## 10. Conclusions

Human papillomavirus is a global public health concern associated with diseases ranging from benign warts to life-threatening malignancy. Research over the last half-century has provided important insight into the biology of the virus and its clinical impact. The development of a successful vaccine has reduced HPV-related disease and offers hope for use in therapeutics; however, barriers to this vaccine still exist, especially in low- and middle-income countries. Efforts in education, vaccination campaigns, and continued research are needed to reduce HPV-related morbidity and mortality worldwide.

## Figures and Tables

**Figure 1 viruses-16-00680-f001:**
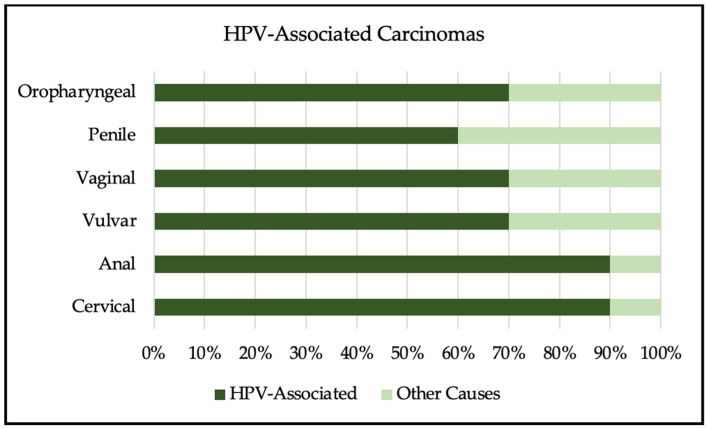
Depiction of HPV-related cancers comparing percentages of HPV-associated cause and other causes [9,10].

**Figure 2 viruses-16-00680-f002:**
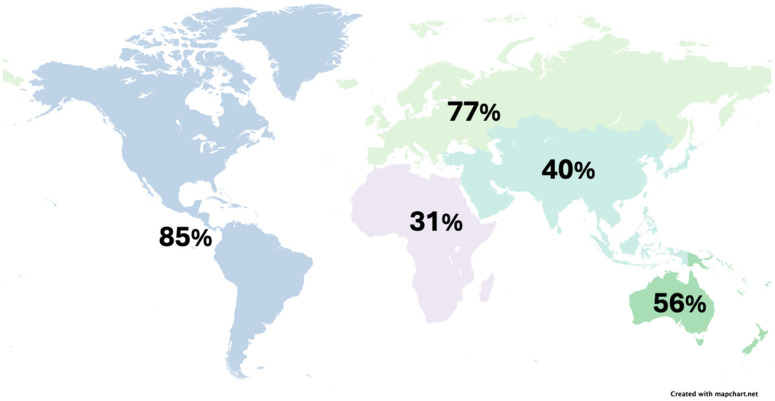
World map representing the percentages of HPV vaccine availability by WHO region: the Americas, Europe, Oceania, Asia, and Africa. Map courtesy of mapchart.net under Creative Commons licensing.

**Table 1 viruses-16-00680-t001:** Cervical cancer and HPV screening guidelines for women by age and HIV co-morbidity based on the WHO, USPSTF, and CDC recommendations [19,20,22].

Population (Female)	USPSTF and CDC	WHO
Under 21	Not recommended	Not recommended
21–30	Pap test every 3 years	Not recommended
30–65	Pap test with HPV co-testing every 5 years, HPV testing every 5 years, or Pap test every 3 years	HPV DNA testing every 5–10 years; complete after 2 negative tests
Living with HIV	Pap test or Pap test with co-testing starting at age 21 yearly for three years then every three years for life	HPV DNA test starting at age 25 every 3–5 years

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
