# Peer review of "Human Papillomavirus and Associated Cancers: A Review"

_viruses, 2024, doi:10.3390/v16050680_

Round 1
Reviewer 1 Report
Comments and Suggestions for Authors
This compact review article focuses on human papillomavirus infection and associated cancers. Although the review article is compact, it is informative, covers all aspects related to the topic and is written in good language.
Minor comments:
Line 218 – The sentence "Human immunodeficiency virus (HIV) is an immunosuppressive disease..." should be corrected because the virus itself is not a disease. Human immunodeficiency virus (HIV) is a virus that attacks the body's immune system by targeting white blood cells. If left untreated, HIV infection can progress to AIDS (Acquired Immune Deficiency Syndrome), which is the most advanced stage of HIV infection.
Line 382 – year of publication is missing.
Author Response
Reviewer 1:
This compact review article focuses on human papillomavirus infection and associated cancers. Although the review article is compact, it is informative, covers all aspects related to the topic and is written in good language.
Minor comments:
Line 218 – The sentence "Human immunodeficiency virus (HIV) is an immunosuppressive disease..." should be corrected because the virus itself is not a disease. Human immunodeficiency virus (HIV) is a virus that attacks the body's immune system by targeting white blood cells. If left untreated, HIV infection can progress to AIDS (Acquired Immune Deficiency Syndrome), which is the most advanced stage of HIV infection.
Reworded in lines 287-291 to state “Human immunodeficiency virus (HIV) is a virus that targets human helper T cells, known as CD4 T cells, and can lead to immunodeficiency as the virus continues to attack T cells [57]. The last, and most severe stage, of HIV infection is acquired immunodeficiency syndrome (AIDS) [58].”
Line 382 – year of publication is missing.
Corrected in Ellingson et al by adding year 2023. Now seen in line 481.

Reviewer 2 Report
Comments and Suggestions for Authors
The review by JaNiese E Jensen et al. although an interesting topic, in my opinion, fails to arouse valid interest on general readers. The review summarizes approximately half a century of research on HPV. Although the concise writing is an advantage, in this case it is unfortunately associated with some incomplete aspects of HPV pathology that would require a more careful and detailed description.
Some examples:
-HPV-related tumor pathology in the anogenital area is substantially different from that of the oral cavity in both biological and clinical terms. The authors should have discussed these aspects.
- Screening is described only as co-testing in USA versus VIA in LMICs, HPV testing alone for screening is not adequately highlighted although used in several countries.
- The use of the preventive vaccine in a "therapeutic" setting should be described in a more precise manner by citing a greater number of works on the topic
- An emerging novelty in the field of HPV-associated tumor pathology is the possibility of viral DNA detection in the plasma of patients with important clinical significance, especially in oropharyngeal tumors. In this review, authors make no mention of this methodology.
Finally, authors should consider using tables and figures to make the review more interesting and effective.
In conclusion, the review could be reconsidered for publication only after profound modifications.
Author Response
Reviewer 2:
The review by JaNiese E Jensen et al. although an interesting topic, in my opinion, fails to arouse valid interest on general readers. The review summarizes approximately half a century of research on HPV. Although the concise writing is an advantage, in this case it is unfortunately associated with some incomplete aspects of HPV pathology that would require a more careful and detailed description.
Some examples:
-HPV-related tumor pathology in the anogenital area is substantially different from that of the oral cavity in both biological and clinical terms. The authors should have discussed these aspects.
Addressed in lines 61-78 with “HPV infection causes cancer due to upregulation of E6 and E7 reading frames in the HPV genome which inactivate p53 and Rb, respectively, in their host cells [11]. The upregulation of E6 and E7 can lead to an abortive infection in which there is dysregulation of the viral genome preventing the production of infectious viruses. HPV can also lead to productive infections that allow for the viral life cycle to function properly. The specific mechanism of carcinogenesis after HPV infection varies based on location of the infection. Cervical cancer commonly develops where the cervix transitions from stratified squamous epithelium to columnar epithelium known as the transformation zone. This area is postulated to consist of reserve cells that act as stem cells. Once infected with HPV, the viral gene expression cannot be regulated leading to metaplasia and eventually cancer. The dysregulation of the viral life cycle in reserve cells is due to an increase in abortive infections rather than productive infections. Anal cancer due to HPV infection occurs in a similar pattern to cervical cancer as HPV often infects the anal transformation zone leading to cancer. Both cervical and anal HPV carcinogenesis differ from the likely mechanism of HPV-associated oropharyngeal cancer. The oropharynx is made up of tonsillar crypt epithelium which more readily allows HPV infection that is more permissive of abortive infections than productive infections. This increases the risk for dysregulated viral gene expression that can lead to cancer.”
- Screening is described only as co-testing in USA versus VIA in LMICs, HPV testing alone for screening is not adequately highlighted although used in several countries.
Addressed in lines 110-121 with “As of 2021, the WHO recommends screening with HPV DNA testing only, rather than Pap test cytology or Visual Inspection with Acetic Acid (VIA), for all women starting at age 30 with an interval of every five to ten years with completion after two negative screening tests [22]. This recommendation changes for women living with HIV. Initial screening with HPV DNA testing starts at age 25 at an interval of three to five years [22]. The comparison of guidelines is shown in Table 1. A study of 12,113 women in The Netherlands found the prevalence of high risk HPV to be 8% when screening with only HPV DNA testing [23]. HPV DNA screening of 2453 Iranian women showed 10.3% had high risk HPV [24]. The WHO notes that HPV DNA screening allows for self-sampling and could be used to increase screening rates as some women may feel more comfortable with self-sampling than with traditional sampling with a provider [22].”
- The use of the preventive vaccine in a "therapeutic" setting should be described in a more precise manner by citing a greater number of works on the topic
Addressed in lines 318-334 with “The therapeutic vaccines used in this study were peptide-based, protein-based, viral-vectored, bacterial-vectored, DNA-based, and cell-based vaccines [66]. The targets of many therapeutic vaccines are E6 and E7 proteins which are the main proteins that influence infection and carcinogenesis [66]. It has been postulated that targeting the E2 protein could be an ideal target for therapeutic vaccines as the E2 protein acts as a repressor of E6 and E7 proteins and could therefore prevent dysregulation of p53 and Rb [67]. The therapeutic vaccines differ from the prophylactic vaccines which are recombinant vaccines from viral-like particles [43]. This meta-analysis found that if a therapeutic vaccine is given to a patient with CIN II or III, the pooled outcome was a 54% regression of the lesions to normal or to CIN I [66]. Comparatively, patients who had CIN II or III and were given a placebo vaccine had a regression of 27% [66]. One study from the meta-analysis looked at a therapeutic vaccine only targeting the E2 protein and showed a complete clearance of lesions in individuals with CIN I, II, or III lesions [68]. There are currently multiple therapeutic vaccines undergoing phase II and III trials, none of which have received approval for use [69]. This meta-analysis shows that using a therapeutic HPV vaccine as treatment for CIN may be feasible and may help lower rates of cervical cancer and associated deaths.”
- An emerging novelty in the field of HPV-associated tumor pathology is the possibility of viral DNA detection in the plasma of patients with important clinical significance, especially in oropharyngeal tumors. In this review, authors make no mention of this methodology.
Addressed in lines 142-151 with “Screening for HPV is primarily focused on patients who are considered high risk. This means screening is often for cervical and anal cancers, with limited screening options for HPV-associated oropharyngeal cancers. A potential solution to this is screening with a plasma circulating marker for HPV known as ctHPVDNA. In a study of 97 patients with locally confined HPV-associated oropharyngeal squamous cell carcinoma, HPV16 ctDNA was detected in 90 patients and HPV33 ctDNA was detected in three patients [29]. This shows that the ctHPVDNA test is 95.6% sensitive and all controls had undetectable levels, therefore a 100% specificity [29]. Although this test is primarily being used in research settings, the high sensitivity and specificity shows promise for use as a screening test, particularly to screen for oropharyngeal cancers related to HPV.”
Finally, authors should consider using tables and figures to make the review more interesting and effective.
Added one table in lines 122-124 and two figures in lines 58-60 and lines 270-273.
In conclusion, the review could be reconsidered for publication only after profound modifications.

Round 2
Reviewer 2 Report
Comments and Suggestions for Authors
The review has been substantially improved, but I believe that the issue “The use of preventive vaccine in a therapeutic context” needs to be discussed more clearly. In other words, there are therapeutic vaccines (now correctly reported in the review) but also preventive vaccines used in a putative therapeutic setting. This last point is a debated issue and the following reviews could help authors discuss it.
1) Reuschenbach M, Doorbar J, Del Pino M, Joura EA, Walker C, Drury R, Rauscher A, Saah AJ. Prophylactic HPV vaccines in patients with HPV-associated diseases and cancer. Vaccine. October 6, 2023;41(42):6194-6205.
2) Jentschke M, Kampers J, Becker J, Sibbertsen P, Hillemanns P. Prophylactic HPV vaccination after conization: a systematic review and meta-analysis. Vaccine. Sep 22, 2020;38(41):6402-6409.
Author Response
Thank you for your suggestion on this point. We have updated the section on therapeutic vaccines and use of prophylactic vaccines as potential treatment for patients already infected with HPV. Please see lines 317-393 for these changes.